# Structural and Photoluminescent Properties of a Novel Terbium Bis(thiocyanato)aurate, Tb[Au(SCN)$_2$]$_3$·6H$_2$O

**Jared D. Taylor and Richard E. Sykora ***

Department of Chemistry, University of South Alabama, 6040 USA Drive South, Chemistry Building Room 223, Mobile, AL 36688, USA; jared.d.taylor@gmail.com
* Correspondence: rsykora@southalabama.edu; Tel.: +1-(251)-460-7422

**Abstract:** The reaction of Tb$^{3+}$ ions with KAu(SCN)$_2$ results in the formation of the crystalline coordination compound Tb[Au(SCN)$_2$]$_3$·6H$_2$O. Single-crystal X-ray diffraction has been employed to investigate the structural features of this compound. The crystallographic data are as follows (Mo K$_\alpha$, $\lambda$ = 0.71073 Å): orthorhombic, *Cmcm*, *a* = 12.4907(9) Å, *b* = 8.5845(6) Å, *c* = 20.7498(8) Å, *V* = 3679.72(16) Å$^3$, *Z* = 4, $R_1$(I > 2($\sigma$)) = 0.0232. This material represents the first known example of a lanthanide dithiocyanatoaurate compound. Au(SCN)$_2^-$ anions bridge Tb$^{3+}$ centers in a bidentate fashion to form the [Tb(H$_2$O)$_4$(Au(SCN)$_2$)$_2$]$_\infty^+$ 1D chains present in the structure. Trimeric Au units in the structure contain short aurophilic bonding interactions with distances of 3.1066(4) Å. The more common O–H···O and O–H···N H-bonding interactions in the structure are overshadowed by relatively rare O–H···S interactions involving the bis(thiocyanato)gold(I) anions. Photoluminescence measurements illustrate that Tb[Au(SCN)$_2$]$_3$·6H$_2$O displays strong Tb$^{3+}$-based emission, but there is a lack of Au-based emission down to 85 K. Excitation spectra are recorded for the title compound and these measurements demonstrate the presence of a donor–acceptor process within the compound, leading to enhanced Tb$^{3+}$-based emission.

**Keywords:** terbium; bis(thiocyanato)Au(I); single-crystal X-ray diffraction; energy transfer; photoluminescence





## 1. Introduction

Lanthanide ions have attractive photophysical properties for many applications, including bioimaging [1], energy conversion [2], and optical communications [3]. These photophysical properties include very narrow line widths, long emissive lifetimes, and high stokes shifts [4]. The transitions that cause this emission are, however, spin-forbidden *f*–*f* electronic transitions, this leads to narrow absorbance bands with poor absorption and thus weak emission. For example, Eu$^{3+}$ and Tb$^{3+}$ are among the most emissive lanthanides and have unsensitized phosphorescence quantum yields of around 10% [5,6]. Despite this fact, these ionic transitions are attractive for many applications. Due to *f*-orbital contraction in lanthanide ions, the emissive orbitals lie beneath the filled 5s and 5p orbitals; thus, they do not participate in bonding and their energy is largely unaffected by the ligand environment [7]. The common approach to improving the intensity of these emissions is through the bonding of chromophores with high absorptivity which can populate the excited lanthanide orbitals through singlet states, triplet states, or charge transfer states, although triplet states seem to be the most prevalent due to relativistic effects [7]. Many antenna complexes have been found that achieve this effect [6], including familiar organic cyclic imine ligands such as 2,2′:6′,2′′-terpyridine (terpy), 2,2′-bipyridine (bpy), and 1,10-phenanthroline (phen) [8]. This is in addition to less familiar inorganic antenna complexes like quantum dots composed of InPZnS or CsPbCl$_3$, which exhibit significant improvements in solid state quantum yields of lanthanide emissions and have broad band absorptivity [4]. Another set of inorganic antenna complexes that have received significant attention and show promise in this field are emissive organometallic Au(I) complexes [9].

The phenomenon of emission in Au(I) chemistry has been heavily explored for decades and has been shown to be a product of the formation of excimers or exciplexes of multiple Au(I) centers based on aurophilic interactions that are similar in strength to hydrogen bonds [10]. The additive strength of this anti-electrostatic interaction is well illustrated by the packing of $[Au(SCN)_2]^-$ anions, where the Au···Au interaction is preferred over the polymerization through the hard–soft interactions of the ligand [11]. Aurophilic interactions are weak supramolecular interactions which are sensitive to many factors, including pressure, temperature, cation identity, doping, and concentration [12]. These factors control the aggregation of Au centers in the solid state as dimers, trimers, tetramers, one-dimensional chains, or a two-dimensional Kagome lattice [11]. Many of these same controls can be extended into the solution state, where photo-oligomerization produces one-dimensional chains which have been confirmed at concentrations as low as $10^{-5}$ M [13]. This control of aggregation is the primary method of tuning the absorption and emission wavelengths of these species and leads to tailorable absorption and emission properties. This is highly desirable as the energy of the donor orbitals can be adjusted to optimize energy transfer to a particular acceptor such as an emissive or reactive center.

Aurophilic interactions as sensitizers for lanthanide emission were first shown by the Patterson group in 2007 when they used the well-studied emissive aurophilic coordination polymer $[Au(CN)_2^-]_n$ and its silver-based congener to demonstrate the concentration and solvent-dependent sensitization of the $Eu^{3+}$ emission intensity [14], eventually showing a 67-fold enhancement in the $Eu^{3+}$ 589 nm emission with the silver complex in solution [9]. The aurophilic emission of $[Au(CN)_2^-]$ seems largely dependent on the length of the exciplex polymer; thus, the tunability of its photophysical properties hinges on having many exciplexes spanning a wide range of chain lengths [13]. Coker, under the supervision of R.C. Elder, demonstrated the emissive behavior of $[Au(SCN)_2^-]$ compounds [15] and later expanded these studies to show it is primarily due to emissive dimers rather than exciplexes of larger nuclearity and further that the emissive energy is directly correlated with Au···Au distance rather than exciplex chain length [16]. Thus, designing a material with several Au···Au dimers containing different distances might afford similar tunability with higher atom efficiency. Coupling this with the flexible, bidentate nature of the thiocyanate ligand could also lead to more efficient energy transfer and less solvent-dependent photophysics. To our knowledge, studies of lanthanide-bearing bis(thiocyanato)Au(I) compounds have not been reported. In efforts to expand knowledge in this area of Au(I) chemistry, herein, we present the structure and photoluminescence studies of a novel terbium coordination polymer, $Tb[Au(SCN)_2]_3 \cdot 6H_2O$.

## 2. Materials and Methods

$Tb(NO_3)_3 \cdot xH_2O$ (Alfa Aesar, Ward Hill, MA, USA, 99.9%), $HAuCl_4 \cdot 3H_2O$ (Alfa Aesar, 99.99%), KSCN (Strem, Newburyport, MA, USA, 99.7%), and tetrahydrothiophene ($C_4H_8S$) (Aldrich, 99%, St. Louis, MO, USA) were used as obtained from the manufacturer, without further purification. The organic solvents used in the syntheses were of ACS grade. The purity of the millimicron-filtered water, which was utilized in all experiments presented here, was established by ensuring the resistance of the water was 18 MΩ·cm or higher. The highest yields of the title compound were observed for the reaction conditions that are provided below. $K[Au(SCN)_2]$ was prepared from $HAuCl_4 \cdot 3H_2O$, $C_4H_8S$, and KSCN as reported in the literature [15,17]. Elemental analysis data were collected using an Agilent 4100 MP-AES (Agilent Technologies, Inc., Santa Clara, CA, USA) fitted with a nebulizer, glass spray chamber, and easy-fit torch. Nitrogen was supplied from an Agilent 4107 Nitrogen Generator with compressed air supplied from an air compressor. An Agilent SPS 3 autosampler was used to deliver samples to the instrument. Terbium and gold values were determined against calibration standards. IR spectra were recorded on a ThermoScientific Nicolet iS50 FT-IR (Thermo Fisher Scientific, Madison, WI, USA). All IR samples that were investigated in these studies were pressed into KBr pellets and maintained at room temperature. The IR spectra were measured over the range of 400 to

4000 cm$^{-1}$. Raman measurements were performed on a home-built spectrometer system containing the following components. Sample excitation was performed with a 532 nm fiber-coupled laser source (Laser Quantum Gem) focused through a Superhead fiber optic probe (Horiba Scientific, Atlanta, GA, USA). The resultant data were collected with a Spectrum One (Horiba Jobin Yvon) CCD detector that contains a liquid-nitrogen-cooled array, after being passed through a Spex 1870 (SPEX Industries, Metuchen, NJ, USA) half meter monochromator. LabVIEW (National Instruments) code that was written in-house was used to control the entire system. Raman measurements were conducted at room temperature on crystalline samples in the range of 100–2700 cm$^{-1}$.

## 2.1. The Synthesis of Tb[Au(SCN)$_2$]$_3$·6H$_2$O

The synthesis of Tb[Au(SCN)$_2$]$_3$·6H$_2$O was carried out by combining 1 mL of a 0.05 M solution of Tb(NO$_3$)$_3$·xH$_2$O (4:1 ratio of MeCN:H$_2$O) and 1 mL of 0.15 M K[Au(SCN)$_2$] (MeCN). The solvent was allowed to evaporate slowly for a period of several days. After this time, colorless crystals with an 85% yield were recovered. Elemental analysis for Tb[Au(SCN)$_2$]$_3$·6H$_2$O provided an experimental Tb:Au ratio of 1:2.93 (theoretical Tb:Au ratio of 1:3). IR (KBr, cm$^{-1}$): 3420 (s, br), 2137 (s), 2109 (m), 2090 (w, sh), 1622 (m), 1577 (w), 1436 (m), 1117 (m). Raman (neat solid, cm$^{-1}$): 2147 (s), 2130 (s), 2112 (m), 1045 (m), 734 (w), 295 (s), 208 (w).

## 2.2. Single-Crystal X-ray Diffraction Studies

An Xcalibur E single-crystal X-ray diffractometer (Oxford Diffraction Products, Oxfordshire, UK) was used for the structural analysis of the title compound. This instrument is supplemented with a Cryostream 600 series temperature controller (Oxford Cryosystems, Long Hanborough, UK) that provides sample cooling. A stereomicroscope was used for selection of the single crystals prior to mounting on the tips of quartz fibers with epoxy. Subsequently, the crystals were aligned on the instrument by utilizing a digital camera. An Enhance sealed-tube X-ray source (Oxford Diffraction Products) was used to irradiate the crystal with Mo Kα radiation during the experiments. The positions and intensities of diffracted X-rays were measured with a Peltier-cooled Eos area detector (Oxford Diffraction Products). The CrysAlisPro [18] software suite (version 1.171.36.31a) handled instrument control during all phases of the experiment, including pre-experiment crystal selection and full data collection strategy and control. CrysAlisPro also contains the functionality for the subsequent processing of raw X-ray data, including unit cell determination and refinement, integration the intensities of the reflection data, space group determination, and implementation of absorption corrections.

Following data collection and preliminary processing, files were generated and further processed by the suite of programs included in SHELX [19]. Specifically, SHELX was used for the structure solution (XS) and least-squares refinement (XL) of the structure. Initial structural models were created by using both the Patterson method as well as direct methods; the solution supplied by direct methods was subsequently used along with several cycles of difference maps to locate the remaining atomic positions. Anisotropic displacement parameters were included in the final refinements for all of the terbium, gold, sulfur, carbon, nitrogen, and oxygen sites in the structure. The hydrogen atomic sites were refined isotropically. The final refinement utilized a full-matrix least-squares calculation. Table 1 contains a list of select crystallographic data resulting from the diffraction experiments. Additional crystallographic details, including the cif file, checkcif report, Tables S1–S7, and Figures S1–S4, are included in the Supplementary Materials. CSD 2258115 also contains the supplementary crystallographic data for this paper. These data can be obtained free of charge as it has been deposited in Access Structures, the joint service of the Cambridge Crystallographic Data Centre and FIZ Karlsruhe for the deposition of crystal structures.

**Table 1.** Crystallographic data for Tb[Au(SCN)$_2$]$_3$·6H$_2$O.

| Formula | C$_6$H$_{12}$Au$_3$N$_6$O$_6$S$_6$Tb |
|---|---|
| Formula weight (amu) | 1206.40 |
| Crystal System | Orthorhombic |
| Space group | *Cmcm* (No. 63) |
| *a* (Å) | 12.4907(9) |
| *b* (Å) | 8.5845(6) |
| *c* (Å) | 20.7498(8) |
| *V* (Å$^3$) | 3679.72(16) |
| Z | 4 |
| T (K) | 180 |
| λ(Å) | 0.71073 |
| $\rho_{calcd}$ (g cm$^{-3}$) | 3.602 |
| μ(Mo *Kα*) (mm$^{-1}$) | 23.458 |
| $R(F_o)$ for $F_o^2 > 2\sigma(F_o^2)$ [a] | 0.0232 |
| $R_w(F_o^2)$ [b] | 0.0572 |

[a] $R(F_o) = \sum||F_o| - |F_c||/\sum|F_o|$. [b] $R_w(F_o^2) = \left[\sum\left[w(F_o^2 - F_c^2)^2\right]/\sum wF_o^4\right]^{1/2}$.

### *2.3. Luminescence Measurements*

A Photon Technology International (PTI) QuantaMaster spectrometer system was utilized for all of the photoluminescence studies performed on the title compound. This instrument contains a 75 W continuous Xe source and photomultiplier tube detectors (Hamamatsu model 928 tube) operating in digital (photon counting) mode to measure steady-state emission and excitation spectra. In addition, time-dependent measurements are made possible by using a pulsed Xe source and gated PMT tubes (Hamamatsu model 928 tube). In the case of either excitation light source, the selection of excitation and emission wavelengths was conducted by means of computer-controlled, autocalibrated "QuadraScopic" monochromators, which are equipped with aberration-corrected emission and excitation optics. The emission monochromators were calibrated using an NIST traceable tungsten light to compensate for the wavelength-dependent variation in the system in the emission channel. The instrument operation, data collection, and handling were all controlled using the FeliX32 fluorescence spectroscopic package. Variable-temperature measurements were made possible with the use of a Janis VNF-XG cryostat system that was controlled with a LakeShore Model 325 temperature controller (LakeShore Cryotronics, Westerville, OH, USA). All of the luminescence experiments were conducted on solid-state, crystalline samples of the title compound held in sealed quartz capillary tubes.

## 3. Results and Discussion

### *3.1. Structural Studies*

The structure of Tb[Au(SCN)$_2$]$_3$·6H$_2$O consists of [Tb(H$_2$O)$_4$(Au(SCN)$_2$)$_2$]$^+$ 1D chains formed from bidentate bridging of [Au(SCN)$_2$]$^-$ anions between two neighboring Tb$^{3+}$ cations. Figure 1 provides a thermal ellipsoid plot with an atom labeling scheme that shows the coordination environments of the metal centers. The coordination of the sole crystallographic Tb site is eight-fold and can be described as a slightly distorted [TbO$_4$N$_4$] square antiprism. The four N atoms in the inner sphere of the Tb site all reside on one square face of the antiprism, likewise for the remaining four water molecules. The Au centers are coordinated by *S*-bound thiocyanate ligands in two-coordinate, linear coordination environments. Previous structural studies have shown the torsion angle is highly variable for [Au(SCN)$_2$]$^-$ anions. They have been observed in different structural arrangements, including staggered, eclipsed, and multiple intermediate conformations [20–22]. Tb[Au(SCN)$_2$]$_3$·6H$_2$O exhibits two crystallographically distinct [Au(SCN)$_2$]$^-$ anions, with torsion angles of 180° and 97.9°.

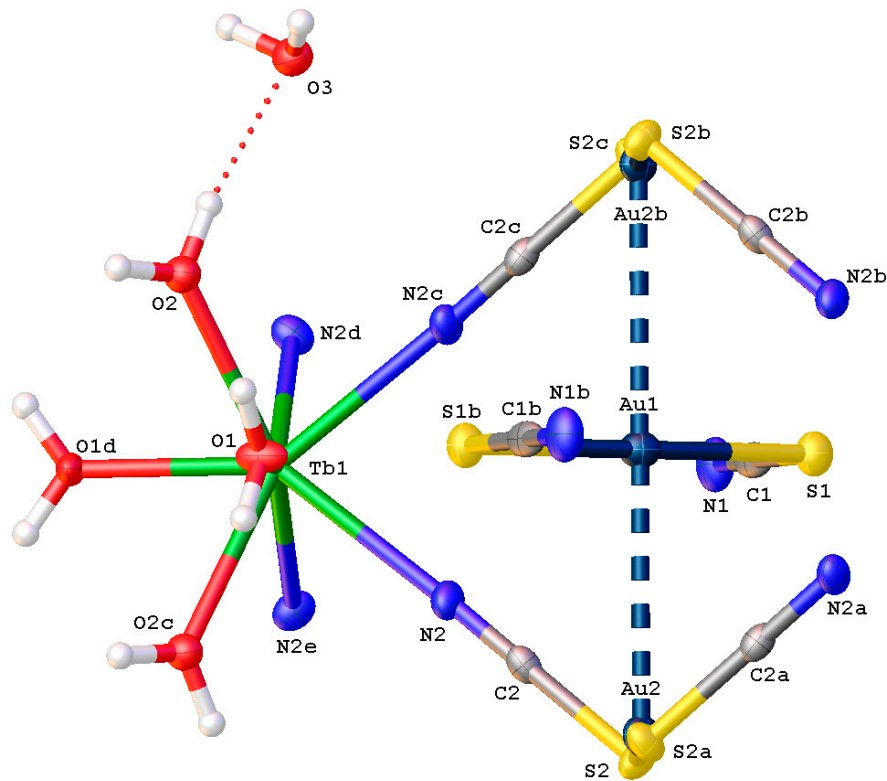

**Figure 1.** A thermal ellipsoid plot (50%) shows the local coordination environment of each metal and the symmetry contained in the structure of Tb[Au(SCN)$_2$]$_3$·6H$_2$O. Symmetry transformation used to generate equivalent atoms: (a) x, 1 − y, 1 − z; (b) 1 − x, 1 − y, 1 − z; (c) 1 − x, y, z; (d) 1 − x, y, $\frac{1}{2}$ − z; (e) x, y, $\frac{1}{2}$ − z.

The chain structure that results due to the bridging of Tb$^{3+}$ by the bis(thiocyanato)gold(I) anions is illustrated in Figure 2. Bidentate coordination of the gauche [Au(SCN)$_2$]$^−$ anions is responsible for the chain formation, while uncoordinated, staggered bis(thiocyanato)gold(I) anions do not ligate the Tb$^{3+}$ site. The two types of anions are involved in aurophilic bonding with each other in the structure; the uncoordinated anions reside between two coordinated bis(thiocyanato)gold(I) ligands. Gold(I) chemistry is filled with examples of aurophilic interactions that produce 2D sheets [23], 1D chains [24], or smaller oligomeric units [25]. To be considered an important aurophilic interaction, Au···Au separations should be approximately 3.5 Å or less, while many interactions are under 3 Å [26]. The aurophilic interactions in Tb[Au(SCN)$_2$]$_3$·6H$_2$O are 3.107 Å and result in the formation of isolated trimeric units within the structure. The next shortest Au to Au separation is 4.292 Å, much too long to be identified as an important bonding feature in this compound. The aurophilic interactions within the related dicyanoaurate compound, Tb[Au(CN)$_2$]$_3$·3H$_2$O [27], on the other hand are more than 0.2 Å longer and exist in the form of 2D networks of Kagome sheets. In addition to the strong aurophilic interactions in the title compound, an extensive hydrogen bonding network, illustrated in Figure 3, also adds to its structural stability. Traditional hydrogen bonding involving the coordinated waters, lattice water, and terminal N of the bis(thiocyanato)aurate anions is present. However, a rarer type of O–H···S hydrogen bonding is exhibited in this structure. Though longer relative to traditional hydrogen bonds involving N or O acceptors, hydrogen bonding with sulfur has been described to be very important in biological systems [28,29] and the O–H···S hydrogen bond length of 2.55 Å in this compound is in line with previous examples. For example, the hydrogen bond values are quite similar to those reported previously for the O–H···S and N–H···S hydrogen bonding exhibited in thiocyanate structures [30]. A complete listing of H-bond parameters for Tb[Au(CN)$_2$]$_3$·3H$_2$O can be found in the Supporting Materials, Table S7.

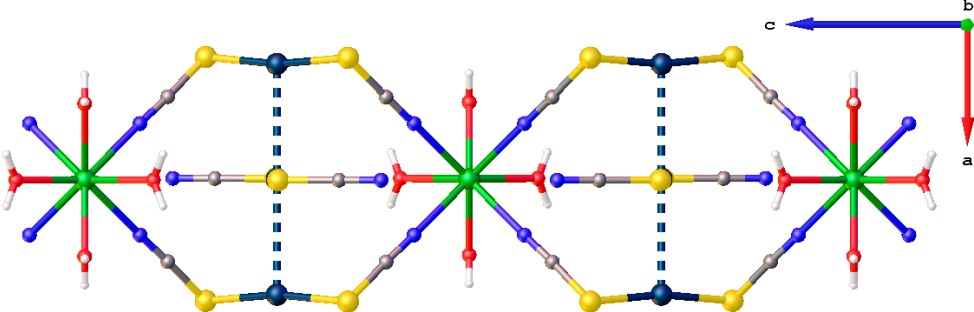

**Figure 2.** A ball and stick plot illustrates the 1D cationic $[Tb(H_2O)_4(Au(SCN)_2)_2]_\infty^+$ chains present in Tb[Au(SCN)_2]_3·6H_2O along with the uncoordinated dithiocyanatoaurate anions.

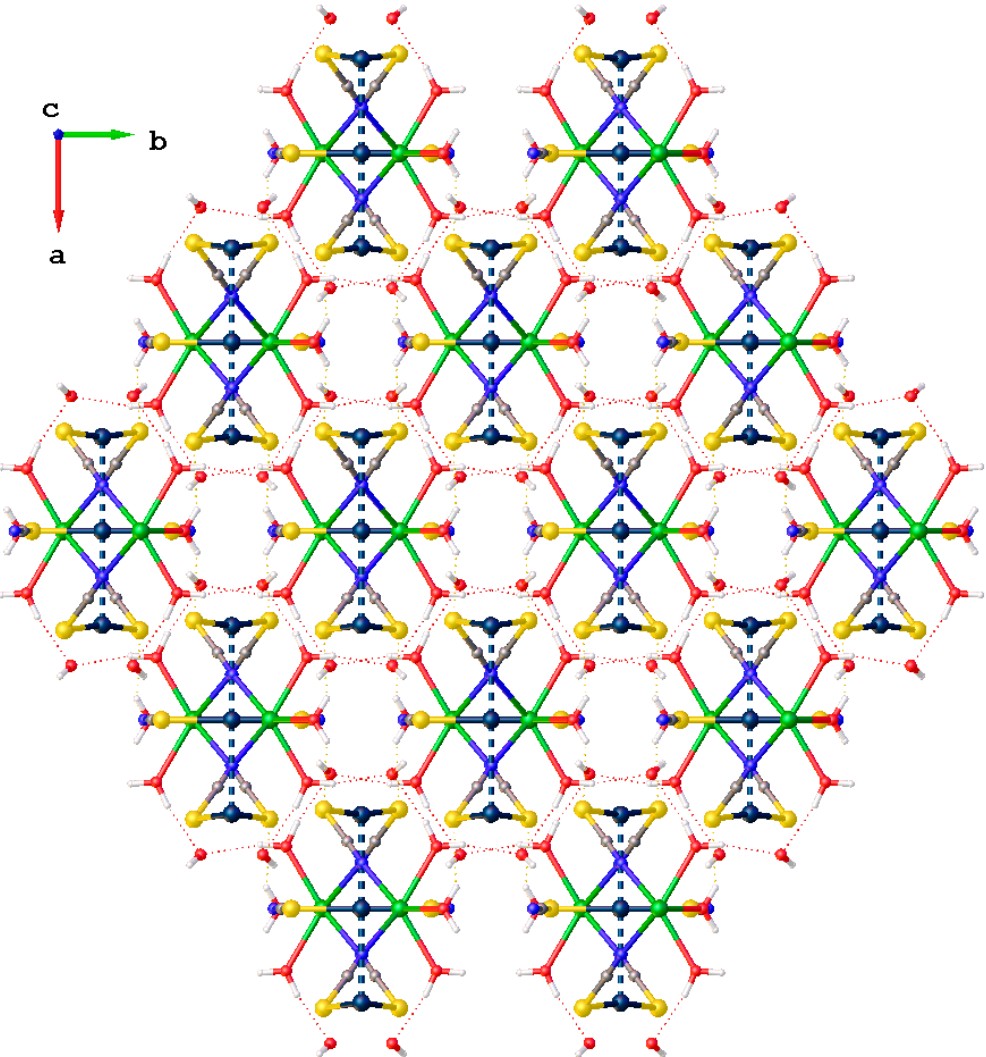

**Figure 3.** A packing diagram for Tb[Au(SCN)_2]_3·6H_2O, viewed down the *c* axis, that illustrates the packing of the one-dimensional chains.

The bond distances for Tb[Au(SCN)_2]_3·6H_2O can be found in Table 2, and a comprehensive list of all bond lengths and angles is included in the Supporting Materials. A clear trend in the Tb–O and Tb–N bond distances between the two coordinated ligands is observed, as the Tb–O distances are shorter by approximately a tenth of an Angstrom relative to the Tb–N bonds. The Au–S, S–C, and C–N distances each vary little, with no observable trend in regard to terminal versus bridging origins. The values of bond

distances in previously reported structures are consistent with all of the values observed herein [15,22,27].

**Table 2.** Important bonding distances (Å) for $Tb[Au(SCN)_2]_3 \cdot 6H_2O$.

| Distances (Å) | | | |
|---|---|---|---|
| Tb1–N2 (x4) | 2.455(6) | C1–N1 | 1.169(11) |
| Tb1–O1 (x2) | 2.347(6) | C2–N2 | 1.153(8) |
| Tb1–O2 (x2) | 2.389(7) | Au1–S1 | 2.318(3) |
| S1–C1 | 1.669(9) | Au2–S2 | 2.2974(15) |
| S2–C2 | 1.673(7) | Au1–Au2 | 3.1066(4) |

*3.2. Luminescence Measurements*

The temperature-dependent emission spectra of $Tb[Au(SCN)_2]_3 \cdot 6H_2O$ upon excitation at 350 nm contain a series of sharp and strong bands located within the visible region. Representative spectra covering the range of temperatures studied are provided in Figure 4. Table 3 contains the band locations and assignments for the spectrum recorded at 175 K. Upon lowering the temperature from room temperature to 175 K, the relative intensity of the emission (which can be assigned as $Tb^{3+}$-based) increases; however, it then decreases upon additional cooling to 85 K. This dip in intensity can be easily understood based upon the excitation features presented shortly. The assignment of the sharp, strong bands in the emission spectrum is straightforward since these bands are characteristic of the $^5D_4 \longrightarrow {}^7F_J$ (J = 0–6) transitions of the $Tb^{3+}$ ion [31]. The emission lifetime measured at 544 nm has a value of 419.6 µs, also consistent with this assignment as originating from $Tb^{3+}$ $f$–$f$ transitions. In related $Tb^{3+}$ compounds containing noble-metal-based ligands, $Tb^{3+}$ has been reported to have lifetimes ranging from 330 to 760 µs [12,32,33]; therefore, the value reported for the title compound is well within this range of values. Similar emission profiles, albeit with a greatly reduced intensity, are apparent from direct $Tb^{3+}$ excitation. For example, excitation at 488 nm, corresponding to the $^5D_4 \leftarrow {}^7F_6$ transition, provides relatively weak Tb-based emission due to low absorption of the $f$-element ion. In previously reported dithiocyanatoaurates, strong Au-based emission has been reported as occurring in the visible region [15,16]. Therefore, a series of temperature- and wavelength-dependent emission experiments covering the ranges from 85 to 293 K and 200 to 488 nm was performed. However, nowhere in this range of temperatures, nor in this range of excitation wavelengths, is the appearance of any emission bands corresponding to Au-based ligand emission observed.

**Table 3.** Band locations and assignments for the emission transitions of $Tb[Au(SCN)_2]_3 \cdot 6H_2O$ measured at 175 K.

| Assignment | λ (nm) |
|---|---|
| $^5D_4 \longrightarrow {}^7F_6$ | 487,491 |
| $^5D_4 \longrightarrow {}^7F_5$ | 543,549 |
| $^5D_4 \longrightarrow {}^7F_4$ | 581,583,589 |
| $^5D_4 \longrightarrow {}^7F_3$ | 617,621 |
| $^5D_4 \longrightarrow {}^7F_2$ | 644,653 |
| $^5D_4 \longrightarrow {}^7F_1$ | 667,671 |
| $^5D_4 \longrightarrow {}^7F_0$ | 679 |

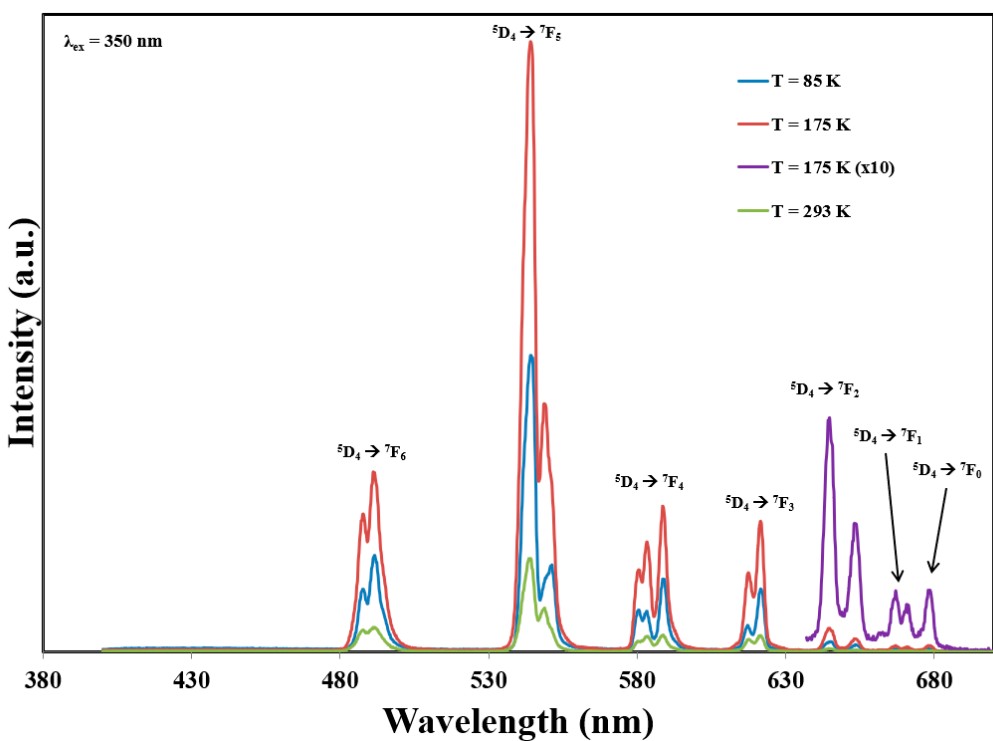

**Figure 4.** Variable-temperature solid-state emission spectra for Tb[Au(SCN)$_2$]$_3$·6H$_2$O.

Figure 5 displays the temperature dependency of the excitation spectra measured at 543 nm and the position of the most intense emission band ($^5D_4 \longrightarrow {}^7F_5$). The spectrum at each temperature is dominated by one strong, broad band in the UV. In addition, a series of sharp and weaker bands are present with maxima at 359, 370, 377, and 487 nm. The weak, sharp bands at lower energy are representative of the Tb$^{3+}$ ion $^5D_j \longleftarrow {}^7F_6$ absorption bands, while the broad band at higher energy can be attributed to Au-based excitation. Figure 4 demonstrates that excitation of this broad band leads solely to characteristic Tb$^{3+}$ *f–f* emission bands, down to at least 85 K. This observation is strong evidence of energy transfer in this compound, where the sensitized emission is realized as a result of energy transfer from the Au-based ligand system to the acceptor Tb$^{3+}$ ion. The Au-based excitation band steadily increases in intensity, narrows, and blue shifts upon cooling (peak maximum of 350 nm at 293 K and 330 nm at 85 K). This band shift provides the rationale for the reduced emission intensity upon cooling. As additional evidence, a series of temperature-dependent emission spectra were measured, for which the excitation wavelength was steadily reduced to correspond to the excitation maximum. Figure S7 shows this series of spectra, which confirms the trend of increasing emission intensity as the temperature decreases. On the other hand, the Tb$^{3+}$ excitation bands show a relatively small temperature dependence, as seen in Figure 5.

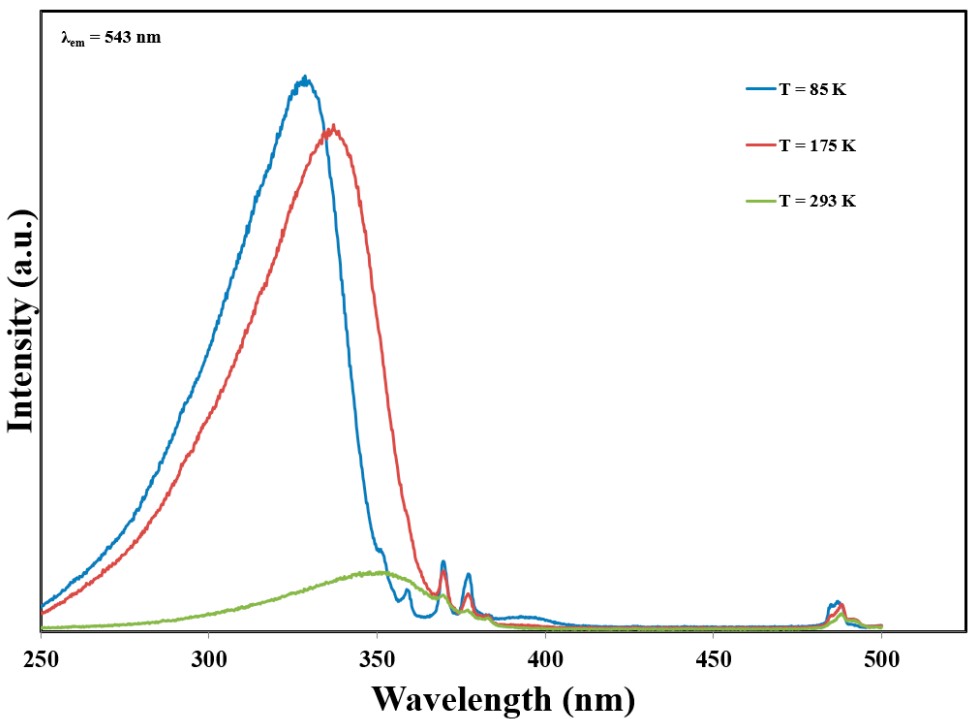

**Figure 5.** Variable-temperature solid-state excitation spectra for $Tb[Au(SCN)_2]_3 \cdot 6H_2O$.

## 4. Conclusions

Structural and temperature-dependent luminescence spectroscopy was conducted on $Tb[Au(SCN)_2]_3 \cdot 6H_2O$, the first structurally characterized lanthanide dithiocyanoaurate compound. The structural analysis reveals that this compound contains a one-dimensional structure containing cationic $[Tb(H_2O)_4(Au(SCN)_2)_2]_\infty^+$ chains exhibiting bidentate, bridging $[Au(SCN)_2]^-$ anions co-crystallized with additional uncoordinated $[Au(SCN)_2]^-$ anions. An additional prominent structural feature is the presence of trimeric Au moieties via aurophilic interactions. $Tb[Au(SCN)_2]_3 \cdot 6H_2O$ lacks Au-based emission down to 85 K, but exhibits a strong green emission characteristic of $Tb^{3+}$ emission. Donor–acceptor energy transfer is apparent within the compound, which leads to an enhancement in the lanthanide-based emission.

**Supplementary Materials:** The following supporting information can be downloaded at https://www.mdpi.com/article/10.3390/inorganics11110419/s1: additional structural details from the single crystal X-ray diffraction experiment, (checkcif report, cif file, Figures S1–S4; Tables S1–S7), IR and Raman spectra (Figures S5 and S6), and photoluminescence spectra (Figure S7).

**Author Contributions:** Conceptualization, R.E.S.; methodology, J.D.T. and R.E.S.; software, J.D.T. and R.E.S.; validation, J.D.T. and R.E.S.; formal analysis, J.D.T. and R.E.S.; investigation, J.D.T. and R.E.S.; resources, R.E.S.; data curation, J.D.T.; writing—original draft preparation, J.D.T. and R.E.S.; writing—review and editing, J.D.T. and R.E.S.; visualization, R.E.S.; supervision, R.E.S.; project administration, R.E.S.; funding acquisition, R.E.S. All authors have read and agreed to the published version of the manuscript.

**Funding:** This research did not receive any external funding. The sole source of monies to promote these studies was internal funding from the University of South Alabama Department of Chemistry.

**Data Availability Statement:** Supplementary crystallographic data have been deposited with Access Structures, the joint service of the CCDC and FIZ Karlsruhe, and can be obtained free of charge from https://www.ccdc.cam.ac.uk/structures/by requesting CSD 2258115 (accessed on 23 October 2023).

**Acknowledgments:** The authors thank Jason W. Coym from the Department of Chemistry at the University of South Alabama for hands-on instruction with the MP-AES.

**Conflicts of Interest:** The authors declare no conflict of interest. The funders had no role in the design of the study; in the collection, analyses, or interpretation of data; in the writing of the manuscript; or in the decision to publish the results.

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
