# Peer review of "Structural and Photoluminescent Properties of a Novel Terbium Bis(thiocyanato)aurate, Tb[Au(SCN)2]3·6H2O"

_inorganics, doi:10.3390/inorganics11110419_

Round 1
Reviewer 1 Report
This is a nice manuscript regarding the luminesent properties of a Tb-Au complex. The work has been clearly presented and, in my opinion, it deserves to be published after minor consideration:
- the authors must indicate if the luminescence studies were conducted in solid state or in solution;
- this lack of trend between the emission intensity and temperature must be tried to discuss/rationalize
- the effect of the Au complex on the recorded emission can be discussed by comparing the emission energies and/or lifetimes measured with other Tb complexes reported in the literature
Author Response
Responses to Reviewers’ Comments
Reviewer 1.
Recommendation: “This is a nice manuscript regarding the luminesent properties of a Tb-Au complex. The work has been clearly presented and, in my opinion, it deserves to be published after minor consideration”
Author Response: The authors thank this reviewer for the time to review our manuscript and for the kind words regarding the high quality of our manuscript.
Reviewer Comment: “the authors must indicate if the luminescence studies were conducted in solid state or in solution”
Author Response: The luminescence studies were conducted on solid state samples. On page 4, lines 136-137 (Section 2.3 Photoluminescence Studies) of the original manuscript submission the following sentence described the nature of the luminescence samples. “All of the spectroscopic experiments are conducted on neat crystalline samples held in sealed quartz capillary tubes.” We have revised the wording of this sentence in the revised manuscript in an attempt to clarify.
Reviewer Comment: “this lack of trend between the emission intensity and temperature must be tried to discuss/rationalize”
Author Response: We have elaborated on the trend between the emission intensity and temperature in the revised manuscript in order to clarify the relationship. In particular, the temperature-dependent shift of the excitation band was reinvestigated and emission spectra corresponding to the maximum were recorded. These new spectra are included in the Supporting Information as Figure S7.
Reviewer Comment: “the effect of the Au complex on the recorded emission can be discussed by comparing the emission energies and/or lifetimes measured with other Tb complexes reported in the literature”
Author Response: We have include a comparison of the lifetime of the title compound with that of previously reported Tb complexes.
Reviewer 2 Report
The paper by Taylor and Sykora represents the synthesis and characterization of bimetallic terbium-gold complex Tb[Au(SCN)2]3•6H2O. The compound obtained represents the first known example of a lanthanide dithiocyanatoaurate compound. The crystal structure of the complex and its photoluminescence are presented. The research is routine but adds additional clues to the chemistry of lanthanides. The research is suitable for publication in Inorganics, but could be accepted only after careful revisions:
1. What does mean the terminus “fluorescence” in the introduction (l. 29-31)? Complexes of Lanthanides demonstrate phosphorescence (the same is presented in refs. 5-6)
2. The complex has not been practically characterized. The elemental analysis or X-ray powder analysis (compared to Single-crystal) demonstrating the purity of the full array should be given. The relations of bands in IR and Raman spectra also need to be presented (for example, the presence of C-N starching vibrations, O-H). Why are there several bands in the region corresponding to C-N vibrations in IR spectra? Spectra should be given in the supporting information.
3. The cif file is also suggested to be given in SI. Surprisingly but there is only checkcif. Table 1 is not required in the main text
4. The photoluminescence part arises questions. The authors talk about the dependence of excitation and emission on temperature. But, the PL spectra were measured only at 350 nm excitation. The excitation maxima extremely depend on the temperature based on Figure 5. For example, the emission at 85 K corresponds to the low-intense 5Dj-7F6 absorptions, not to gold-based transitions demonstrating maximum at ca. 320-330 nm. In contrast, there is some capturing of au-based excitation at 175 K. In addition, it is very difficult to use the same conditions for measurements of solid samples in contrast to solutions and make discussions about relative intensities (em/ex slits, the position of the sample etc.). Please comment on the procedure of measurements. The authors should present the emissions at different excitation maximums including higher energy bands (ca. 300 nm) at corresponding temperatures. For example, according to ref .15, the emissions of bis(thiocyanato)gold(I) mainly have been measured at 320 nm excitations. What about the dependence of lifetime on the temperature?
Author Response
Responses to Reviewers’ Comments
Reviewer 2.
Recommendation: “The research is suitable for publication in Inorganics, but could be accepted only after careful revisions”
Author Response: The authors thank this reviewer for the time to review our manuscript and for the recommendation to publish the work. We have made changes and additions to the manuscript in response to this reviewer’s detailed comments. The sections below describe these in detail.
Reviewer Comment: “1. What does mean the terminus “fluorescence” in the introduction (l. 29-31)? Complexes of Lanthanides demonstrate phosphorescence (the same is presented in refs. 5-6)”
Author Response: We thank the reviewer for catching this embarrassing error. We have replaced the word fluorescence with phosphorescence.
Reviewer Comment: “2. The complex has not been practically characterized. The elemental analysis or X-ray powder analysis (compared to Single-crystal) demonstrating the purity of the full array should be given. The relations of bands in IR and Raman spectra also need to be presented (for example, the presence of C-N starching vibrations, O-H). Why are there several bands in the region corresponding to C-N vibrations in IR spectra? Spectra should be given in the supporting information.”
Author Response: We have included elemental analysis for the title compound in our revised submission as requested by this reviewer. The authors don’t agree with the general comment that “the complex has not been practically characterized.” Our original submission included two forms of vibrational spectroscopy (IR and Raman) in addition to a detailed structural investigation with SCXRD, as well as the temperature-dependent photoemission spectroscopy. The vibrational spectra have been added to the Supporting Information. The crystal structure indicates multiple independent cyanide groups in the structure, which matches well the splitting observed for the CN vibrations in both the IR and Raman spectra. A full group theory analysis of these vibrational modes has not been conducted as part of this work.
Reviewer Comment: “3. The cif file is also suggested to be given in SI. Surprisingly but there is only checkcif. Table 1 is not required in the main text”
Author Response: The cif file was deposited with the Cambridge Database prior to manuscript submission and the deposition number included in the manuscript. In the revised submission we have included a copy of both the cif file and checkcif report to be included as supplementary materials. Following the manuscript review, the editorial staff commented that the manuscript was too short and asked that we lengthen the manuscript. For that reason, we have added many details to the manuscript to increase the length and for that same reason did not remove Table 1.
Reviewer Comment: “4. The photoluminescence part arises questions. The authors talk about the dependence of excitation and emission on temperature. But, the PL spectra were measured only at 350 nm excitation. The excitation maxima extremely depend on the temperature based on Figure 5. For example, the emission at 85 K corresponds to the low-intense 5Dj-7F6 absorptions, not to gold-based transitions demonstrating maximum at ca. 320-330 nm. In contrast, there is some capturing of au-based excitation at 175 K. In addition, it is very difficult to use the same conditions for measurements of solid samples in contrast to solutions and make discussions about relative intensities (em/ex slits, the position of the sample etc.). Please comment on the procedure of measurements. The authors should present the emissions at different excitation maximums including higher energy bands (ca. 300 nm) at corresponding temperatures. For example, according to ref .15, the emissions of bis(thiocyanato)gold(I) mainly have been measured at 320 nm excitations. What about the dependence of lifetime on the temperature?”
Author Response: We have conducted a new set of temperature-dependent luminescence experiments and that data is included in the Supporting Information, Figure S7. As this reviewer so appropriately pointed out, the temperature dependence of the ligand band is very temperature-dependent. For that reason, we tuned the excitation for the emission at each temperature to correspond to the maximum excitation. The new set of data more clearly show the trend in emission that we had referred to in the original manuscript.
In regards to the reviewer’s comment about the challenges of handling solid samples, the authors strongly agree. It is critical to minimize the number of variables in these types of experiments. For that reason, we have utilized constant lamp and slit settings during the course of our temperature dependent measurements. Also, we have not moved or repositioned the sample in any way following the initial mounting within our cryostat. Lastly, we unfortunately do not the capability to conduct variable-temperature lifetime measurements. The PTI instrument that contains our cryostat does not contain this functionality.
Round 2
Reviewer 2 Report
The paper could be published in its present form